# Performance Study of a Piezoelectric Energy Harvester Based on Rotating Wheel Vibration

**DOI:** 10.3390/mi17010006

**Published:** 2025-12-20

**Authors:** Rui Wang, Zhouman Jiang, Xiang Li, Xiaochao Tian, Xia Liu, Bo Jiang

**Affiliations:** 1School of Engineering, Changchun Normal University, Changchun 130032, China; 2Rare Metal Intensive Processing Engineering Research Center of Jilin Province, Changchun Normal University, Changchun 130032, China; 3School of Mechanical and Vehicle Engineering, Changchun University, Changchun 130022, China

**Keywords:** piezoelectric energy harvester, wheel vibration, synchronous switching circuit, piezoelectric vibrators

## Abstract

To address the issue of low efficiency in recovering low-frequency vibration energy during vehicle operation, this paper proposes a piezoelectric energy capture harvester based on wheel vibration. The device employs a parallel configuration of dual cantilever beam piezoelectric transducers in its mechanical structure, with additional mass blocks to optimize its resonant characteristics in the low-frequency range. A synchronous switch energy harvesting circuit was designed. By actively synchronizing the switch with the peak output voltage of the piezoelectric element, it effectively circumvents the turn-on voltage threshold limitations of diodes in bridge rectifier circuits, thereby enhancing energy conversion efficiency. A dynamic model of this device was established, and multiphysics simulation analysis was conducted using COMSOL-Multiphysics to investigate the modal characteristics, stress distribution, and output performance of the energy harvester. This revealed the influence of the piezoelectric vibrator’s thickness ratio and the mass block’s weight on its power generation capabilities. Experimental results indicate that under 20 Hz, 12 V sinusoidal excitation, the system achieves an average output power of 3.019 mW with an average open-circuit voltage reaching 16.70 V. Under simulated road test conditions at 70 km/h, the output voltage remained stable at 6.86 V, validating its feasibility in real-world applications. This study presents an efficient and reliable solution for self-powering in-vehicle wireless sensors and low-power electronic devices through mechatronic co-design.

## 1. Introduction

In recent years, as automotive wireless sensor networks have evolved toward low-power operation, stable self-powered technologies have become a critical issue in intelligent vehicles. Piezoelectric materials possess electromechanical coupling properties that directly convert mechanical energy into electrical energy, making piezoelectric energy harvesting a significant research direction in micro-energy fields [1,2,3,4,5]. Compared to electromagnetic and thermoelectric energy harvesting methods, piezoelectric energy harvesters offer advantages such as simple structure, ease of miniaturization, strong environmental adaptability, and abundant vibration sources, demonstrating promising application prospects in automotive and various engineering fields [6,7,8,9,10,11].

In automotive applications, low-frequency vibrations induced by radial runout and structural deformation during wheel rotation provide a potential excitation source for energy harvesting [12,13]. Consequently, research on energy harvesting from wheel low-frequency vibrations holds promise for providing continuous power supply to in-vehicle sensors such as TPMS [14,15,16]. However, traditional piezoelectric energy harvesters face challenges in this operating condition, including insufficient low-frequency structural response and significant threshold losses in diode bridge rectifier circuits, which limit overall energy conversion efficiency.

To enhance harvesting performance, researchers have conducted extensive explorations in both structural design and energy management circuits, including multilayer flexible structures, broadband piezoelectric composites, and active power management technologies, significantly advancing the efficiency of piezoelectric energy harvesting [17,18,19,20,21]. These studies indicate that achieving synergistic optimization of structures and circuits under low-frequency wheel vibration environments is a key direction for improving the performance of piezoelectric energy harvesters.

Given the aforementioned challenges, this paper proposes a dual cantilever beam parallel piezoelectric energy harvester integrated with the tire valve core, aiming to significantly enhance the harvestability of low-frequency vibration energy from wheels. By attaching mass blocks to the free ends of both cantilevers and deeply integrating the optimized mechanical structure with an active power management circuit based on synchronous switching energy extraction technology onto a compact PCB, the threshold voltage drop issue inherent in traditional rectifier circuits is effectively overcome, fundamentally improving energy conversion efficiency.

Combined with the designs of other schemes in Table 1, this research demonstrates significant innovation in enhancing low-frequency response, achieving compact structural integration, suppressing circuit losses, and improving system usability. It provides a more engineering-feasible solution for low-power self-powered systems in automotive applications [22,23,24].

To validate the effectiveness of the proposed design, this paper employs COMSOL multiphysics simulation to conduct modal analysis, stress–strain distribution analysis, and transient voltage output analysis. It also investigates the impact of the mass-to-thickness ratio of the piezoelectric vibrator on energy output performance. Based on these findings, a prototype was fabricated and an experimental platform was established. By simulating tire vibration excitation at different vehicle speeds, high-precision output voltage, current, and power waveforms were obtained. Experimental results validate the superior performance of the proposed electromechanical co-optimization strategy under low-frequency conditions, demonstrating the practical application potential of this energy harvester in vehicle self-powered systems.

## 2. Structural Design and Working Principle

The overall structure of the piezoelectric energy capturer is shown in Figure 1, which is mainly composed of five parts: brass substrate, piezoelectric ceramic sheet, mounting shell, mass block, and PCB board. The piezoelectric ceramic is pasted on a brass substrate, and the fixed end of the brass substrate is solidly attached to the mounting housing with bolts and nuts to form a cantilever beam-type piezoelectric vibrator. The mass block is glued to the free end of the brass substrate with epoxy resin. Due to the limited amount of power generated by a single chip piezoelectric vibrator, two piezoelectric vibrators are connected in parallel to increase the power generation.

The piezoelectric effect can be categorized into two modes,
$d_{31}$ and
$d_{33}$, depending on the direction of transformation. In
$d_{31}$-mode, the directions of the external force and the electric field generated by the piezoelectric vibrator are perpendicular to each other. In
$d_{33}$-mode, the electric field generated by the external piezoelectric vibrator is aligned in the same direction. Although the electromechanical coupling coefficient of
$d_{31}$-mode is relatively small, the piezoelectric vibrator can obtain a large strain when a small external force is applied, so it is suitable for applications with low vibration frequency, small external force, and small piezoelectric vibrators. According to the positive piezoelectric effect, when an external force is applied to the free end of a piezoelectric vibrator, causing it to bend and deform, an electric charge is generated on its surface.

In this paper, two single-crystal chip piezoelectric vibrators are connected in parallel as an energy conversion medium. In order to reduce the resonance frequency of the piezoelectric vibrator and amplify the strain during its operation, a mass block is glued to its free end. The rims vibrate up and down when the tires are deformed during the driving of the car, and because the mass block has a large inertia, the inertial force causes the piezoelectric vibrator to bend and deform. The piezoelectric energy harvester utilizes the positive piezoelectric effect of the piezoelectric vibrator to rectify, filter, and store the generated energy in a capacitor for collection through the PCB board.

As shown in Figure 2, the integrated mounting method of the piezoelectric energy capturer and wheel valve core can effectively prevent damage to the internal structure of the tire when it is dismantled. It also has the advantage of taking up little space with its compact structure. The blue part is a schematic diagram of the piezoelectric harvester, which is connected to the valve core on the wheel rim with a nut. The piezoelectric harvester is designed to have a slight curvature to adapt to the stepped structure of the wheel hub itself.

With the fixed end of the piezoelectric vibrator as the moving point, the ground as the fixed reference system, and the rim as the moving reference system, the rim is observed to move in a circular motion. The velocity and acceleration of the circular motion of the wheel rim were analyzed as shown in Figure 3.

In Figure 3, *O* is the center point of the rim, *R* is the radius of the rim,
A is the equivalent point of the fixed end of the piezoelectric vibrator,
w is the angular velocity of the rim, and the point velocity and acceleration are obtained from the synthesis theorem of point *A*, as follows:
(1)$$v_{A}=v_{O}+v_{AO}=v_{O}+Rw$$
(2)$$a_{A}=a_{O}+a_{AO}=a_{O}+{a^{n}}_{AO}+{a^{\tau }}_{AO}=a_{O}+R\frac{dw}{dt}+Rw^{2}$$

After deriving the velocity and acceleration at point
A, the velocity and acceleration at the free end of the piezoelectric vibrator were analyzed. The piezoelectric vibrator motion was analyzed as shown in Figure 4.

Let
Q be the center point of the mass block at the free end of the piezoelectric vibrator, and determine the velocity at point *Q*:
(3)$$v_{Q}=v_{r}+v_{e}=r\times w_{\theta }+v_{e}$$ where
$v_{Q}$ is the velocity at point *Q*,
$v_{e}$ is the traction velocity,
$v_{r}$ is the relative velocity,
r is the spacing from the fixed end of the piezoelectric vibrator to the center of the mass block, and
$w_{\theta }$ is the angular velocity of the piezoelectric vibrator’s vibration.

According to the acceleration synthesis theorem for points,
(4)$$\begin{array}{ll} a_{a} & =a_{r}+a_{e}+a_{c}=a_{r}^{n}+a_{r}^{\tau }+a_{e}^{n}+a_{e}^{\tau }+a_{c}\\ & =r\times w_{\theta }^{2}+r\frac{dw_{\theta }}{dt}+\frac{v_{e}^{2}}{r}+\frac{dv_{e}}{dt}+2w_{\theta }\times v_{r} \end{array}$$ where
$a_{a}$ is the absolute acceleration at point *Q*,
$a_{e}$ is the traction acceleration,
$a_{r}$ is the relative acceleration, and
$a_{c}$ is the Kurtosis acceleration.

Therefore, excitation force *F* is applied to the mass:
(5)$$F=ma_{a}$$ where *m* is the mass block weight.

The analysis shows that the directions
$v_{r}$ and
$a_{r}^{\tau }$ are perpendicular to the free end of the piezoelectric vibrator, which contributes to the deformation of the piezoelectric vibrator. The larger
$v_{r}$ and
$a_{r}^{\tau }$ are, the larger the strain produced by the piezoelectric vibrator.

The length of the piezoelectric vibrator is
L, the width is
b, the thickness of the piezoelectric vibrator is
h, and the thickness of the substrate is
c, where
L >>
h,
c.

Under the action of excitation force
F, the piezoelectric vibrator produces a charge of [23]
(6)$$Q=-\frac{3\phi \lambda (1-\phi +\phi \lambda )g_{31}L^{2}}{\beta_{33}^{T}B{\left(h+c\right)}^{2}}F,$$ where
ϕ=c/(c+h) is the ratio of the thickness of the substrate *c* to the total thickness of the piezoelectric vibrator
(c+h),
$\lambda =E_{m}/E_{p}$ is the ratio of the Young’s modulus of the substrate to that of the ceramic wafer,
$g_{31}$ is the piezoelectric voltage constant,
$\beta_{33}^{T}=1/\varepsilon_{33}^{T}$ is the dielectric isolation ratio,
$\varepsilon_{33}^{T}=1300\varepsilon_{0}$ is the dielectric constant in the direction of vibration, and
$\varepsilon_{0}$ is the vacuum dielectric constant.
$$B=A(1-\phi +\phi \lambda )(1+k_{31}^{2})-3\phi^{2}(1-\phi )\lambda^{2}k_{31}^{2},\;A=\phi^{4}{\left(1-\lambda \right)}^{2}-2\phi (2\phi^{2}-3\phi +2)(1-\lambda )+1,\;k_{31}^{2}=E_{p}g_{31}/\beta_{33}^{T}.$$

The capacitance of the piezoelectric energy harvester is obtained from the electric field equation
Q=CV and Equation (6) as follows:
(7)$$C=\frac{(1-\phi +\phi \lambda )AbL}{\left(1-\phi )\beta_{33}^{T}B(c+h\right)}$$

From Equations (6) and (7), the voltage generated by the piezoelectric energy harvester is obtained as follows:
(8)$$V=-\frac{2\times 3\phi \lambda (1-\phi )g_{31}L}{Ab(c+h)}F,$$

Through the above theoretical analysis, it can be seen that the power generation capacity of the device is directly proportional to the deformation produced by the piezoelectric vibrator, and placing the piezoelectric vibrator horizontally with the wheel hub allows the centrifugal force to induce greater deformation in the piezoelectric vibrator, helping to improve the power generation capacity of the device.

## 3. Analysis of Synchronous Switch Energy Harvesting Circuits and Energy Storage Systems

Traditional piezoelectric energy harvesting circuits commonly employ passive bridge rectification. However, at low excitation levels, the diode’s turn-on voltage threshold significantly reduces the circuit’s energy conversion efficiency. To overcome this limitation, this study designed a synchronous switch energy harvesting circuit, whose system schematic is shown in Figure 5. The core concept of this circuit is to replace the traditional passive diode with an active switch synchronized with the voltage of the piezoelectric element, thereby achieving efficient charge transport and energy accumulation.

The operating principle of this circuit can be divided into two stages:

Energy Accumulation Stage: When the piezoelectric element outputs a forward voltage, the upper diode D1 conducts, charging the intermediate energy storage capacitor
$C_{r}$. Simultaneously, the lower controlled switch S1 remains in the open state.

Energy Transfer Phase: When the control logic module detects that the piezoelectric element’s output voltage has reached its peak, it closes switch S1. At this point, the energy stored in the piezoelectric element is rapidly transferred through diode D2 below to the energy storage capacitor
$C_{L}$ at the load end, thereby supplying power to the downstream circuit.

To perform a quantitative analysis of this circuit’s performance, an energy transfer model was established. Assume the equivalent model of the piezoelectric element consists of a sinusoidal current source
$I_{p}\left(t\right)$ in parallel with its internal capacitance
$C_{p}$. During one vibration cycle, the open-circuit voltage generated by the piezoelectric element is
(9)$$\;V_{p}\left(t\right)=V_{peak}sin(\omega t)$$

During the energy transfer phase, switch S1 closes when
$V_{p}\left(t\right)$ reaches the forward peak voltage
$V_{peak}$. At this point, charge transfers from
$C_{p}$ to
$C_{L}$ until the voltage
$V_{p}\left(t\right)$ across
$C_{L}$ equals the voltage across
$V_{L}$. Neglecting the forward voltage drop
$V_{f}$ across diode D2, the charge transfer process obeys the law of charge conservation.

The total energy of the system before transfer is the sum of the energies stored in
$C_{p}$ and
$C_{L}$:
(10)$$E_{\mathrm{before}}=\frac{1}{2}C_{p}V_{\mathrm{peak}}^{2}+\frac{1}{2}C_{L}V_{L}^{2}$$

After the switch closes and reaches a steady state, the voltages across
$C_{p}$ and
$C_{L}$ are equal, denoted as
$V_{\mathrm{final}}$. According to the law of conservation of charge,
(11)$$C_{p}V_{\mathrm{peak}}+C_{L}V_{L}=(C_{p}+C_{L})V_{\mathrm{final}}$$

The final voltage after transfer can be solved as follows:
(12)$$V_{\mathrm{final}}=\frac{C_{p}V_{\mathrm{peak}}+C_{L}V_{L}}{C_{p}+C_{L}}$$

At this moment, the total energy of the system is
(13)$$E_{\mathrm{after}}=\frac{1}{2}(C_{p}+C_{L})V_{\mathrm{final}}^{2}$$

Therefore, during a single switching operation, the energy dissipated due to charge transfer
$\Delta E_{\mathrm{loss}}$ is
(14)$$\Delta E_{\mathrm{loss}}=E_{\mathrm{before}}-E_{\mathrm{after}}=\frac{C_{p}C_{L}}{2(C_{p}+C_{L})}{\left(V_{\mathrm{peak}}-V_{L}\right)}^{2}$$

In a single switching event, the net energy increment
$\Delta E_{\mathrm{harvest}}$ transferred from the piezoelectric element to the energy storage capacitor
$C_{L}$ represents the change in energy:
(15)$$\Delta E_{\mathrm{harvest}}=\frac{1}{2}C_{L}(V_{\mathrm{final}}^{2}-V_{L}^{2})$$

If the system operates in a vibration environment with a frequency of
f, then, theoretically, the average power collected by the circuit
$P_{\mathrm{harvest}}$ is
(16)$$P_{\mathrm{harvest}}=\Delta E_{\mathrm{harvest}}\cdot f$$

Use the collected power
$P_{\mathrm{harvest}}$ to charge the energy storage capacitor
$C_{L}$. Assuming
$P_{\mathrm{harvest}}$ remains constant during charging, the relationship between the voltage
$V_{L}$ across the terminal
$C_{L}$ and time
t can be derived from the following equation:
(17)$$P_{\mathrm{harvest}}=\frac{dE_{L}}{dt}=\frac{d}{dt}\left(\frac{1}{2}C_{L}V_{L}^{2}\right)=C_{L}V_{L}\frac{dV_{L}}{dt}$$

Separating variables and integrating yields the time required to charge the capacitor
$C_{L}$ from the initial voltage
$V_{\mathrm{initial}}$ to the target voltage
$V_{\mathrm{target}}$:
(18)$$\begin{array}{ll} t_{\mathrm{charge}} & =\int_{V_{\mathrm{initial}}}^{V_{\mathrm{target}}}\frac{C_{L}V_{L}}{P_{\mathrm{harvest}}}dV_{L}\\ & =\frac{C_{L}}{2P_{\mathrm{harvest}}}(V_{\mathrm{target}}^{2}-V_{\mathrm{initial}}^{2}) \end{array}$$

To validate the effectiveness of the aforementioned theoretical model, a set of typical simulation parameters was established for computational analysis. Assume the piezoelectric element vibrates at a frequency of
f=100 Hz, with an equivalent capacitance of
$C_{p}=10\;\mathrm{nF}$, a maximum open-circuit voltage that can be generated of
$V_{\mathrm{peak}}=20\;V$, and a backend energy storage capacitor of
$C_{L}=10\;\mu F$. Assume the starting voltage of the charging process is
$V_{L}=3\;V$. According to the formula, the system voltage
$V_{\mathrm{final}}$ after the switch closes is
(19)$$\begin{array}{ll} V_{\mathrm{final}} & =\frac{\left(10\times 10^{-9})\cdot (20)+(10\times 10^{-6})\cdot (3\right)}{10\times 10^{-9}+10\times 10^{-6}}\\ & \approx 3.017\;V \end{array}$$

The energy transferred per cycle
$\Delta E_{\mathrm{harvest}}$ is
(20)$$\Delta E_{\mathrm{harvest}}=\frac{1}{2}(10\times 10^{-6})\cdot (3.017^{2}-3^{2})\approx 0.51\;\mu J$$

Theoretical power collection
$P_{\mathrm{harvest}}$ is
(21)$$P_{\mathrm{harvest}}=0.51\mu J\cdot 100\mathrm{Hz}=51\;\mu W$$

To charge the energy storage capacitor
$C_{L}$ from
0 V to
5 V, the time required
$t_{\mathrm{charge}}$ is
(22)$$t_{\mathrm{charge}}=\frac{10\times 10^{-6}}{2\cdot (51\times 10^{-6})}(5^{2}-0^{2})\approx 2.45\;s$$

Simulation results indicate that this synchronous switching circuit can efficiently harvest energy from low-voltage piezoelectric elements and charge the energy storage capacitor to a voltage level sufficient for most low-power electronic devices within seconds.

To quantitatively evaluate the performance advantages of the synchronous switch energy harvesting circuit (SSH) over the traditional bridge rectifier circuit, this paper conducted direct comparative experiments on both energy management circuits under identical mechanical excitation and measurement conditions. The experimental system configuration is shown in Figure 6.

As shown in Figure 6, the energy flow originates from the sinusoidal excitation source, passes through the dual cantilever piezoelectric energy harvester, then traverses the switchable full-bridge rectifier circuit and SSHI circuit, ultimately converging into the energy storage capacitor. By selecting the circuit switch, the energy harvester’s output can be connected to either a traditional bridge rectifier circuit (1N4148 diode) or a synchronous switch energy harvesting (SSH) active power management circuit, without altering any mechanical excitation conditions. This switching method ensures that both circuits can operate under identical input conditions, eliminating uncertainties arising from variations in mechanical excitation during repeated experiments.

The method for calculating instantaneous energy in energy storage capacitors is as follows:
$E(t)=\frac{1}{2}CV{\left(t\right)}^{2}$. Under steady-state operating conditions, the output voltage and output power of the two circuits were recorded separately, with the results summarized in Table 2.

It can be seen that under identical excitation conditions, the output voltage of the SSH circuit is approximately twice that of the bridge rectifier, with output power increasing by about 218%, demonstrating a significant advantage in energy extraction.

To further compare the energy storage efficiency of the two circuits, this paper recorded the charging curve showing the terminal voltage of the energy storage capacitor over time, as shown in Figure 7.

Experimental results indicate that the energy transfer efficiency of the bridge rectifier circuit was limited due to the forward voltage drop across the diodes. Consequently, the voltage rise rate of the storage capacitor was relatively slow, ultimately stabilizing at approximately 8.2 V. By synchronously switching the capacitor at the peak of the piezoelectric voltage, charge reversal and efficient extraction were achieved. This enabled the storage capacitor voltage to rise to approximately 16.7 V in a shorter time, with a significantly increased energy growth rate. In summary, the SSH circuit demonstrates significant advantages over traditional bridge rectifier configurations in low-frequency excitation scenarios, such as wheel vibration, establishing itself as a key circuit technology for achieving efficient piezoelectric energy harvesting.

## 4. Simulation and Analysis of Piezoelectric Energy Harvester

In order to improve the service life of the piezoelectric energy harvester and to avoid its operation at resonance frequency, eigenfrequency analysis of the piezoelectric energy harvester was carried out using COMSOL Multiphysics 6.2 simulation software, and finite element analysis of the piezoelectric vibrator’s first-to-fourth order modes was obtained, as shown in Figure 8.

As can be seen from Figure 8, the first four modes of the piezoelectric harvester had frequencies of 12.60 Hz, 119.09 Hz, 154.84 Hz, and 621.57 Hz. The first-order vibration modes were consistent with the motion of the piezoelectric vibrator, while the second-, third-, and fourth-order modes did not conform to the tendency of piezoelectric vibrator motion, affecting the service life of the piezoelectric harvester.

Secondly, a sinusoidal excitation signal with a frequency of 20 Hz and an amplitude of 12 V was applied to the piezoelectric harvester with an attached mass block of 10 g and a piezoelectric vibrator specific thickness ratio of 0.4 to study the stress–strain characteristics and the voltage output characteristics of the piezoelectric harvester in the working process. The simulation results are shown in Figure 9 and Figure 10.

As can be seen from Figure 9, in 0~0.2 s, the maximum stress generated by the piezoelectric harvester was 30.6 MPa, which is less than the permissible stress of the selected materials of the equipment. This indicates that the piezoelectric harvester is in a safe range during normal operation and will not produce the phenomenon of stress yielding and damage fracture. The average output displacement of the piezoelectric harvester was 1.09 mm, and the average output voltage was 13.49 V in 0–1 s. The average output voltage of the piezoelectric harvester was 13.49 V. The displacement and output voltage of the piezoelectric harvester reached the maximum values of 1.24 mm and 14.9 V, respectively, at 0.58 s. The piezoelectric harvester generated the best power when piezoelectric vibrator deformation was maximized.

To further enhance the energy conversion efficiency of piezoelectric energy harvesters in low-frequency vibration environments, this study conducted an extended-range systematic optimization investigation targeting the critical design parameters within the structure: the thickness ratio of the piezoelectric vibrator and the mass of the free-end mass block. The parameter ranges for this simulation test were as follows:

Thickness ratio of piezoelectric transducers:
$$0.30\leq t_{r}\leq 0.50$$

Free-end mass block:
5 g≤m≤15 g

Since single-objective or single-variable scans struggle to fully reflect their comprehensive impact, response surface optimization methods were introduced in this study. First, a two-parameter sweep simulation for parameter combination
$\left(t_{r},\;m\right)$ was performed in COMSOL Multiphysics, extracting the following key response quantities: first-order modal frequency, maximum equivalent strain of the cantilever beam, and peak piezoelectric output voltage.

Based on the simulation results, a second-order polynomial response surface model was constructed:
$$V_{\mathrm{out}}=f(t_{r},m)$$

By analyzing the contour distribution of the response surface and the location of the stationary point, one can intuitively assess the trend of parameter effects on output performance and identify potential regions of global optimal parameters. The comprehensive multi-objective optimization results indicate that when the mass block mass was approximately 8–10 g and the thickness ratio was about 0.35–0.4, the energy harvester achieved maximum output voltage and energy conversion efficiency under low-frequency excitation conditions. The data is shown in Table 3. The simulation results above were used to analyze the piezoelectric vibrator thickness ratio and load mass, with the results shown in Figure 11.

From Figure 11, it can be seen that the output voltage of the piezoelectric captor tended to increase with the increase in the additional mass block mass and piezoelectric vibrator thickness ratio, which is consistent with the previous theory. When the thickness ratio of the piezoelectric vibrator was 0.4 and the mass of the mass block was 10 g, the maximum output voltage of the piezoelectric energy harvester was 21.59 V. By optimizing the thickness ratio of the piezoelectric vibrator and the weight of the mass block, a mass block with a piezoelectric vibrator thickness ratio of 0.4 and a mass of 10 g was selected as a reference for subsequent experiments.

## 5. Experiments and Tests

In order to explore the power generation performance of the piezoelectric energy harvester, a test prototype was fabricated and an experimental test platform was built. The prototype used PLA-FC3D printing material as the raw material for processing the mounting housing, in which the material of piezoelectric vibrator was PZT-5H, and the size parameter was
50 mm×23 mm×0.2 mm

The experimental equipment mainly consisted of a signal generator (Shio SA-PA010, manufactured by Shio Technology Co., Ltd., Wuxi, China), power amplifier (Shio SA-SG030, manufactured by Shio Technology Co., Ltd., Wuxi, China), oscilloscope (RIGOL-DS1102, manufactured by RIGOL Precision Electronics Technology Co., Ltd., Suzhou, China), shaker (Shio SA-JZ005T, manufactured by Shio Technology Co., Ltd., Wuxi, China), piezoelectric energy prototype, and optical precision vibration isolation platform. The experimental system is shown in Figure 12.

The piezoelectric energy harvester was fixed on the shaker. The signal generator output a sinusoidal excitation signal with a frequency of 20 Hz and an amplitude of 12 V, which was processed by the power amplifier and inputted to the shaker. The output was connected to an oscilloscope to measure the output voltage of the piezoelectric vibrator. The piezoelectric vibrator with 5 g, 7 g, and 10 g mass blocks attached was tested experimentally for power generation capability, respectively. The output voltage, output current, and output power are shown in Figure 13. The power generation capacity of piezoelectric energy harvesters attached to blocks of different masses is shown in Table 4.

From Figure 13, it can be seen that the output voltage of the piezoelectric captor reached a maximum of 21.82 V, the output current reached a maximum of 0.238 mA, and the output power reached a maximum of 3.99 mW from 0 to 10 s. These results were averaged for experimental rigor and are shown in Table 2. The heavier the mass of the additional mass block, the better the power generation performance of the piezoelectric captor. This is because greater deformation of the piezoelectric vibrator during vibration results in more power generation. This is consistent with the results of the simulation and theoretical analysis.

To quantitatively evaluate the enhancement effect of a dual piezoelectric vibrator parallel configuration on power generation performance, this study conducted comparative experiments under identical mechanical excitation conditions between a single piezoelectric vibrator configuration and a dual piezoelectric vibrator parallel configuration. Under excitation frequency of 20 Hz and excitation amplitude of 12 V, the measured output performance under different configurations is shown in Table 5.

It can be seen that compared to the single-vibrator configuration, the dual-vibrator parallel structure achieved significant improvements in both output voltage and power, with output power increasing by approximately 2–4 times. This validates the effectiveness of the parallel structure in low-frequency vibration environments.

In order to further investigate the effect of different rotational speeds on the power generation performance of the piezoelectric vibrator, a simulation experiment was carried out, as shown in Figure 14. The experimental setup included a motor, transmission, oscilloscope (RIGOL-DS1102, manufactured by RIGOL Precision Electronics Technology Co., Ltd., Suzhou, China), and piezoelectric energy harvester experimental prototype. The piezoelectric energy harvester was mounted on the motor, and the speed of the motor was changed by the transmission and controlled in the range of 10–90 km/h for experimental testing. The effects of different speeds on the output voltage are shown in Figure 15.

As can be seen in Figure 15, the output voltage of the piezoelectric captor increased and then decreased with different vehicle speeds. When the vehicle speed was 70 km/h, its output voltage reached 6.86 V, which is the best performance of power generation at this time. By analyzing the output voltage of the piezoelectric harvester installed in the shaker and motor, an experimental phenomenon was observed. When the shaker was used as the power source, piezoelectric harvester operated at a low frequency, the vibration amplitude was larger, leading to higher piezoelectric vibrator deformation and generating more surface charge; when the motor was used as the power source, the vibration amplitude of the piezoelectric vibrator was smaller, generating less surface charge.

Relate the vehicle’s linear velocity
V(km/h) to the tire’s circumferential fundamental frequency
$f_{\mathrm{rot}}(\mathrm{Hz})$ (i.e., the fundamental excitation frequency caused by each revolution of the tire) and analyze resonance between the natural frequency and excitation frequency to demonstrate the accuracy of the experiment:
(23)$$f_{\mathrm{rot}}=\frac{V_{\mathrm{km}/h}}{3.6L}$$ where L is the tire circumference (m), which can be approximated from the tire specifications as follows:
(24)L≈2πRL≈π(D)

Calculating the frequency
$f_{rot}$ at 70 km/h for common tires using Equations (23) and (24) yielded the results shown in Table 6. Similarly, the relationship between rotational frequency and vehicle speed was plotted as shown in Figure 16.

One of the tire’s natural frequencies was 12.6 Hz, while the fundamental frequency corresponding to 70 km/h was approximately 9.62 Hz, making it difficult to achieve complete alignment. However, actual road surface excitation exhibits a broad spectrum and transient impacts, leading to a broadened excitation spectrum. Consequently, even if there is a certain deviation from the target frequency, the energy absorber will still be significantly activated, as long as sufficient energy is present in the vicinity of the target frequency. As shown in Figure 17, under 70 km/h conditions, the tire–rim acceleration PSD exhibited a distinct peak in the 12.5–12.7 Hz range, precisely coinciding with the first-order modal frequency of the energy absorber. This explains the experimentally observed optimal output performance at 70 km/h.

## 6. Conclusions

To address the technical bottleneck of limited energy harvesting efficiency in automotive wheel systems under low-frequency vibration conditions, this paper proposes a highly efficient piezoelectric energy harvester integrated with the tire valve core. This device employs an electromechanical co-optimization system and utilizes a double cantilever beam parallel configuration to enhance its response capability to low-frequency vibrations. The design integrates an optimized mechanical structure with an active power management circuit based on synchronous switching energy harvesting technology onto a compact PCB board, thereby effectively reducing rectification losses. Structural parameters were optimized through multiphysics simulation and validated through prototype experiments. The experimental results indicate the following:(1)When a 20 Hz sinusoidal excitation signal with an amplitude of 12 V is applied to the piezoelectric energy harvester, the average output voltage reaches 16.90 V, the output current is 0.1819 mA, and the power output is 3.019 mW.(2)At a wheel speed of 70 km/h, the voltage can reach 6.86 V, at which point the generator achieves its optimal power output.

The piezoelectric energy harvester developed by the institute operates within the confined space of a wheel rim, featuring not only a simple structure but also outstanding power generation capabilities. It also offers new insights for future research on vehicle energy harvesting.

## Figures and Tables

**Figure 1 micromachines-17-00006-f001:**
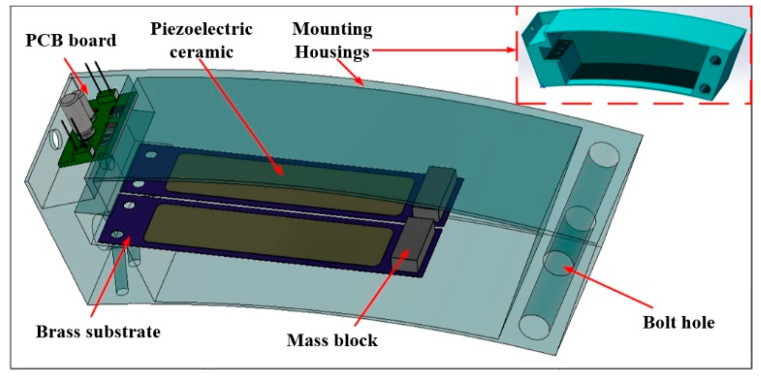
Three-dimensional view of the piezoelectric energy harvester.

**Figure 2 micromachines-17-00006-f002:**
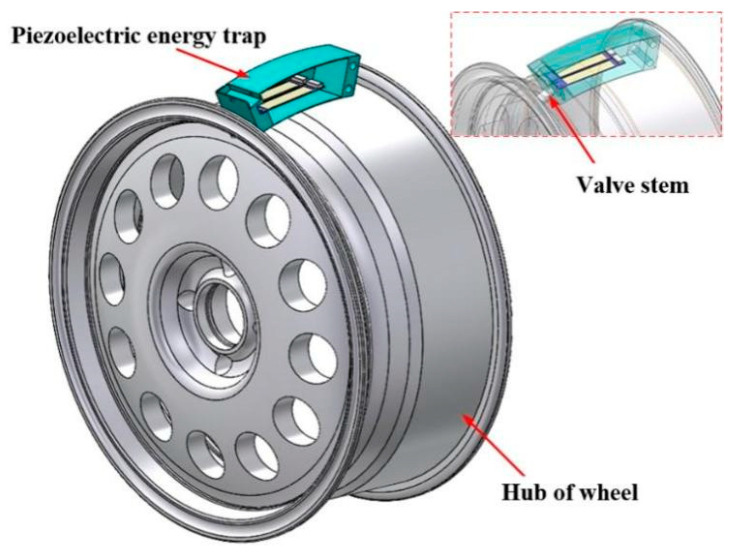
Schematic diagram of the installation position of the piezoelectric energy harvester.

**Figure 3 micromachines-17-00006-f003:**
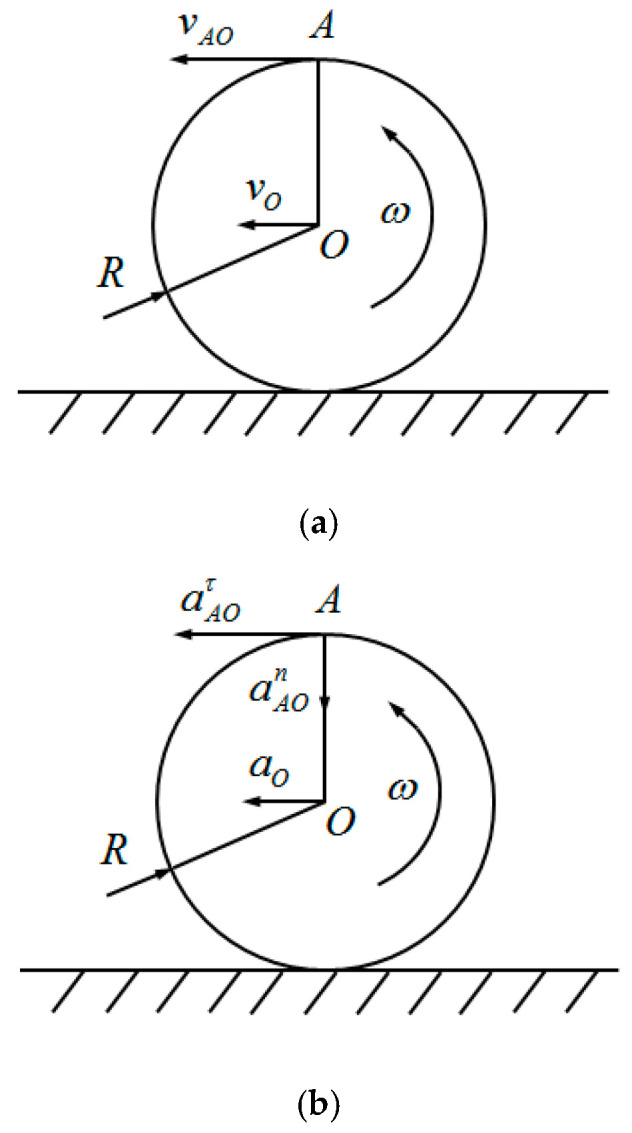
Circumferential motion analysis of washbasins. (**a**) Velocity analysis of washbasins. (**b**) Acceleration analysis of washbasins.

**Figure 4 micromachines-17-00006-f004:**
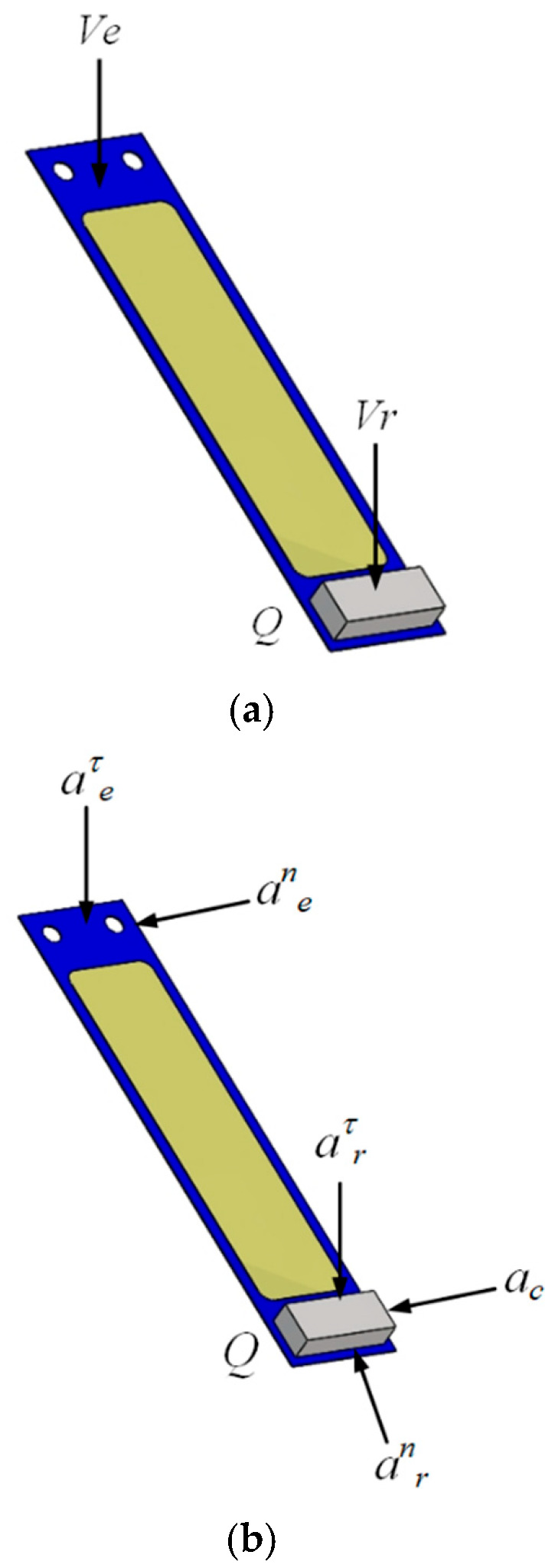
Motion analysis of piezoelectric vibrator. (**a**) Velocity analysis of piezoelectric vibrator. (**b**) Acceleration analysis of piezoelectric vibrator.

**Figure 5 micromachines-17-00006-f005:**
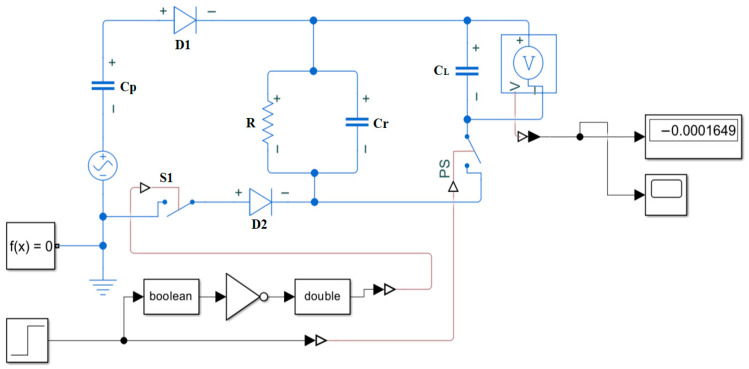
Schematic diagram of synchronous switch energy harvesting circuit.

**Figure 6 micromachines-17-00006-f006:**
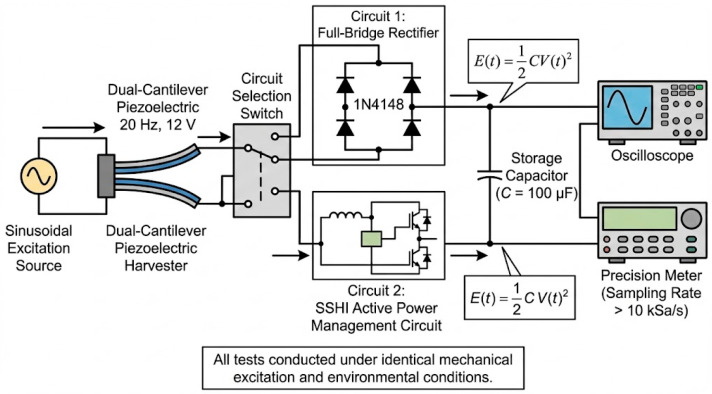
Comparison test diagram of traditional bridge rectifier circuit and synchronous switching energy harvesting circuit.

**Figure 7 micromachines-17-00006-f007:**
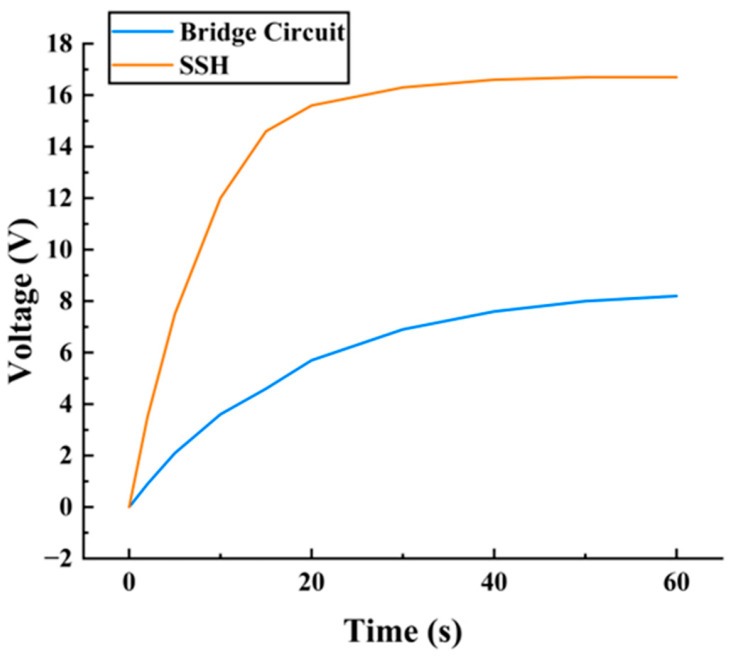
Charging curve showing the terminal voltage of the energy storage capacitor over time.

**Figure 8 micromachines-17-00006-f008:**
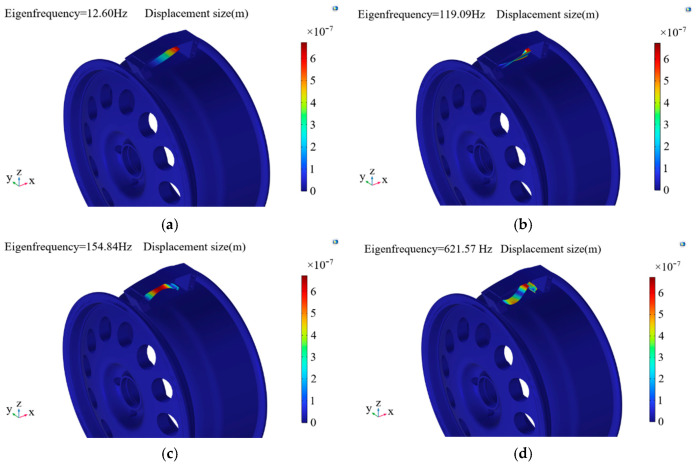
Vibration modes of piezoelectric vibrator. (**a**) First-order modes. (**b**) Second-order modes. (**c**) Third-order modes. (**d**) Fourth-order modes.

**Figure 9 micromachines-17-00006-f009:**
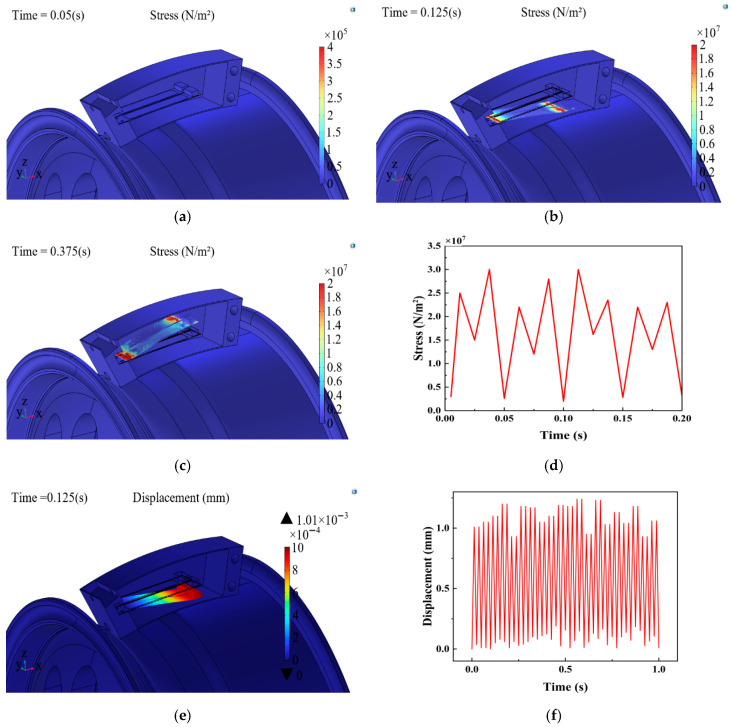
Stress–strain simulation results of piezoelectric energy harvester. (**a**) Stress cloud of piezoelectric energy harvester at 0.05 s. (**b**) Stress cloud of piezoelectric energy harvester at 0.07 s. (**c**) Stress cloud of piezoelectric energy harvester at 0.1 s. (**d**) Stress characteristic curves of the piezoelectric energy harvester at different times. (**e**) Piezoelectric energy harvester strain maps. (**f**) Vibration displacement characteristic curve at different moments.

**Figure 10 micromachines-17-00006-f010:**
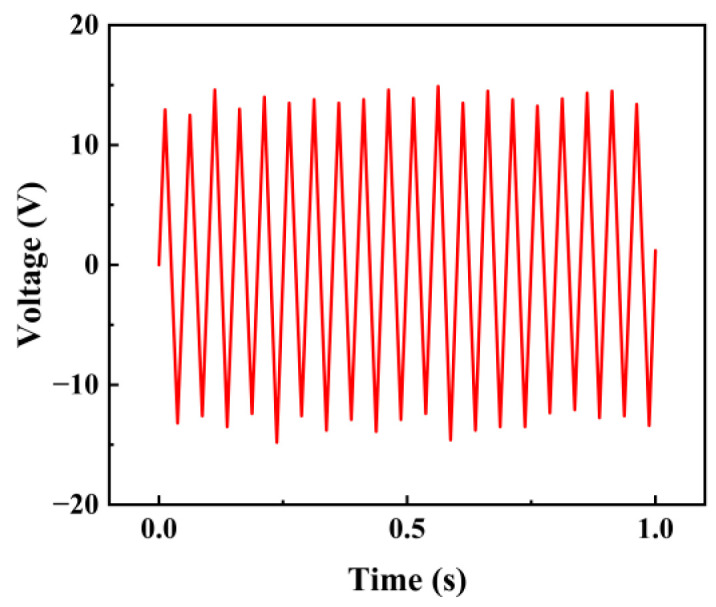
Output voltages of piezoelectric energy harvester at different moments.

**Figure 11 micromachines-17-00006-f011:**
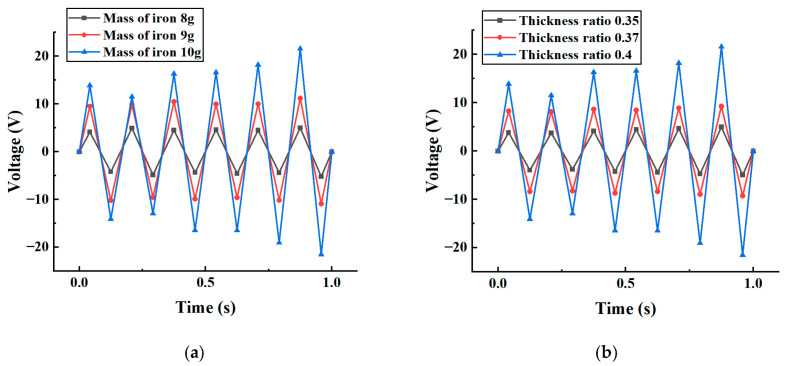
Simulation of output voltages with different mass-to-thickness ratios. (**a**) Plot of output voltages for different masses at a thickness ratio of 0.4. (**b**) Plot of output voltages for different thickness ratios at a mass of 10 g.

**Figure 12 micromachines-17-00006-f012:**
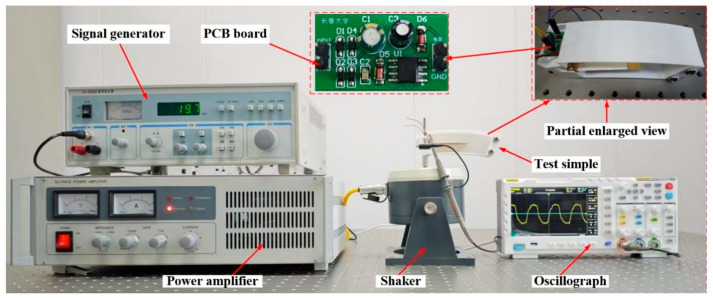
Experimental test platform for piezoelectric captive energy device.

**Figure 13 micromachines-17-00006-f013:**
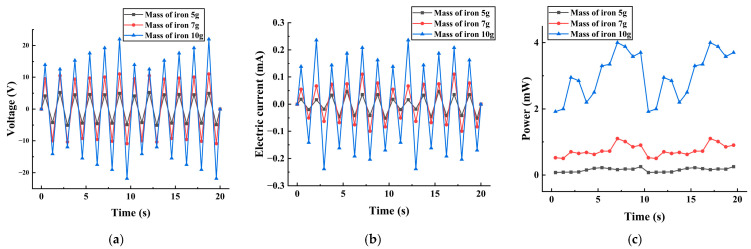
Output voltage, output current, and output power of three sets of piezoelectric energy harvesters. (**a**) Output voltage characteristic curve at different moments. (**b**) Output current characteristic curve at different moments. (**c**) Output power characteristic curve at different moments.

**Figure 14 micromachines-17-00006-f014:**
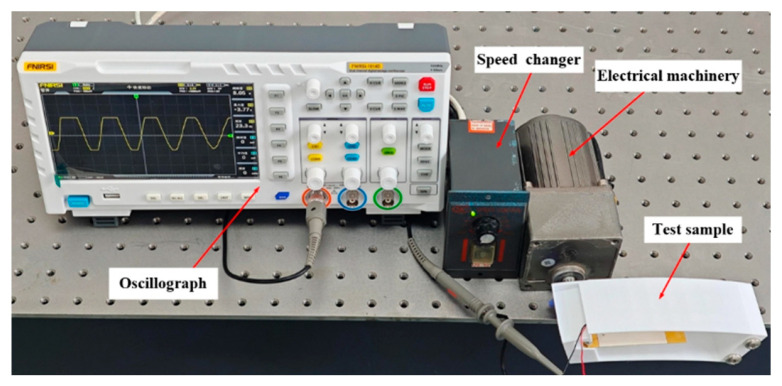
Simulated rotational speed experiment.

**Figure 15 micromachines-17-00006-f015:**
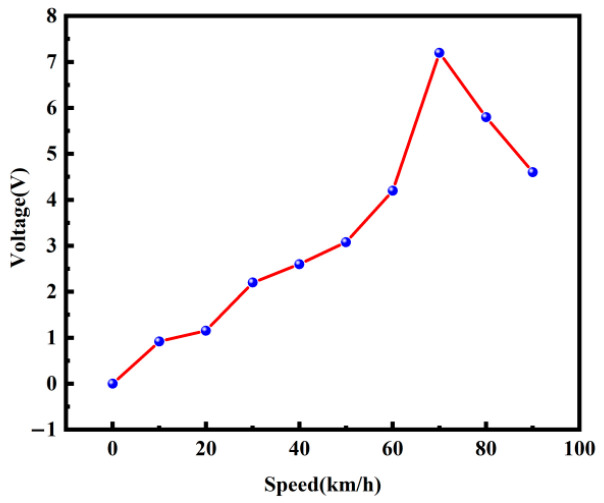
Relationship curve showing the relationship between different speeds and output voltages.

**Figure 16 micromachines-17-00006-f016:**
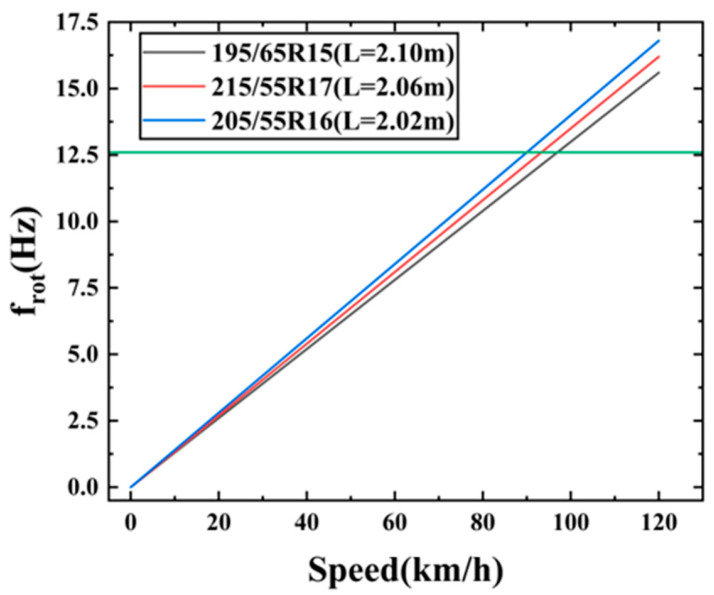
Relationship between rotation frequency and vehicle speed curve.

**Figure 17 micromachines-17-00006-f017:**
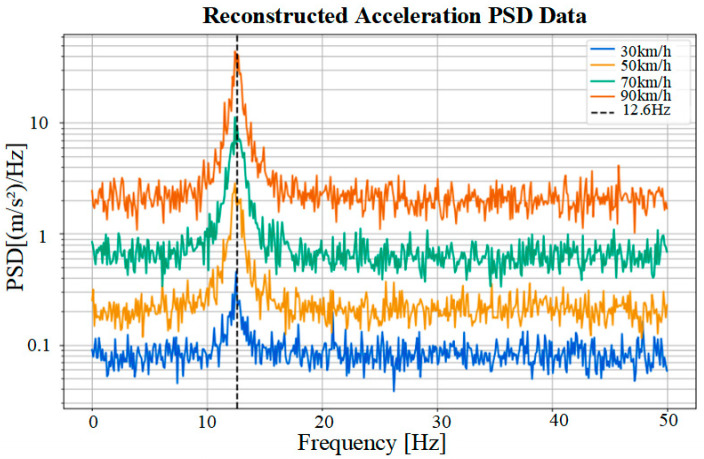
Relationship between acceleration power spectral density at different speeds.

**Table 1 micromachines-17-00006-t001:** Comparison of this study with other studies.

	Structural Type	Application Scenarios	Typical Excitation Frequency	Output (or Performance)	Limitations
H. Staaf et al. [22]	Flexible film patch/tire coupling simulation experiment	Tire/wheel energy harvesting	<20 Hz	Focus on tire deformation modeling and simulation—experimental comparison	Failure to achieve efficient circuit integration
J.R. Leppe-Nerey et al. [23]	PVDF/composite membrane-embedded structure	Integrated within the tire	Low frequency	Material is supple and suitable for extreme deformation	Integration with rigid rims is challenging, and output power is limited
M. Liu et al. [24]	Multi-piezoelectric input synchronous switching topology	Multi-unit parallel connection/grid connection	Low frequency–medium frequency	Supports multi-unit parallel operation with strong scalability	Increasing complexity and implementation costs
This study	Dual cantilever beam parallel configuration, SSH circuit integration	Integrated tire valve stem	10–20 Hz	Significant low-frequency response, compact structure, synchronous switching circuitry reduces rectification losses, high system integration	Engineering durability and long-term fatigue require further validation

**Table 2 micromachines-17-00006-t002:** Performance comparison between traditional bridge rectifier and SSH circuit.

Circuit Type	Output Voltage (V)	Output Power (mW)	Performance Enhancement
Bridge rectifier	8.2	0.95	—
SSH	16.7	3.019	enhance 218% (Power)

**Table 3 micromachines-17-00006-t003:** Mass block and piezoelectric vibrator thickness ratio parameters.

Mass Block	Weight (g)	Piezoelectric Vibrator	Thickness Ratio
Mass block A1	8	Piezoelectric vibrator B1	0.4
Mass block A2	9	Piezoelectric vibrator B2	0.37
Mass block A3	10	Piezoelectric vibrator B3	0.35

**Table 4 micromachines-17-00006-t004:** Generation capacity of piezoelectric energy harvester with different mass blocks attached.

Mass block weight/g	5	7	10
Output voltage/V	4.62	10.03	16.70
Output current/mA	0.03369	0.07437	0.1819
Output power/mW	0.155	0.748	3.019

**Table 5 micromachines-17-00006-t005:** Output performance under different configuration methods.

Layout Method	Output Voltage (V)	Output Current (mA)	Output Power (mW)
Single piezoelectric transducer	8.4	0.095	0.80
Parallel connection of two piezoelectric transducers	16.7	0.1819	3.019

**Table 6 micromachines-17-00006-t006:** Tire circumference conversion chart.

Specifications	Approximate Perimeter L (m)	70 km/h Corresponds to $f_{rot}(\mathrm{Hz})$
205/55R16	2.02	9.62
215/55R17	2.06	9.43
195/65R15	2.10	9.26

## Data Availability

The original contributions presented in this study are included in the article. Further inquiries can be directed to the corresponding author.

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
