# Peer review of "Performance Study of a Piezoelectric Energy Harvester Based on Rotating Wheel Vibration"

_micromachines, 2025, doi:10.3390/mi17010006_

Round 1
Reviewer 1 Report
Comments and Suggestions for Authors
This paper proposes a piezoelectric energy harvester integrated with the tire valve core. This device adopts a parallel configuration of dual cantilever beam piezoelectric transducers, and is equipped with mass blocks to optimize low-frequency resonance characteristics. Combined with a synchronous switching energy harvesting circuit, it overcomes the diode threshold limitation of traditional rectifier circuits, and enhances energy conversion efficiency through mechatronic co-design. However, revisions are needed to address the followed weaknesses. After appropriate revisions to improve the scientific rigor and practical relevance, the manuscript will meet the publication standards of the journal.
- Supplement Literature Comparison and Innovation Positioning: Add a table comparing the proposed harvester with 3–5 representative existing studies (covering structural design, application scenario, excitation frequency, output power/voltage, and advantages). Clearly explain how the dual cantilever parallel structure and synchronous switching circuit address the limitations of previous designs.
- Deepen Mechanism Analysis: Combine the first-order modal frequency (12.60 Hz) obtained from simulations to analyze the relationship between vehicle speed and wheel vibration frequency. Explain why the vibration frequency at 70 km/h matches the harvester’s resonant characteristics, thereby justifying the optimal output performance at this speed.
- Optimize Parameter Optimization and Supplement Circuit Details: Expand the parameter optimization range (e.g., mass block weight: 5–15 g; thickness ratio: 0.3–0.5) and use systematic optimization methods to determine the global optimal parameters.
- Verify Practical Adaptability: Select 1–2 typical low-power in-vehicle devices (e.g., tire pressure sensors) as loads. Conduct experiments to test the harvester’s ability to power these devices and provide specific data to demonstrate practical application value.
Author Response
|
Response to Reviewer 1 Comments
|
||
|
1. Summary |
|
|
|
Thanks very much for your time to review this manuscript. I really appreciate all your comments and suggestions. We have considered these comments carefully and tried our best to address every one of them. In addition, we made minor adjustments to the title of the manuscript to make it more specific and precise, highlighted it in yellow in the marked manuscript file to indicated changes Once again, we appreciate for your warm work earnestly and hope that the corrections will meet with approval. |
||
|
2. Questions for General Evaluation |
Reviewer’s Evaluation |
Response and Revisions |
|
Does the introduction provide sufficient background and include all relevant references? |
Can be improved |
|
|
Are all the cited references relevant to the research? |
Can be improved |
|
|
Is the research design appropriate? |
Can be improved |
|
|
Are the methods adequately described? |
Can be improved |
|
|
Are the results clearly presented? |
Can be improved |
|
|
Are the conclusions supported by the results? |
Can be improved |
|
|
3. Point-by-point response to Comments and Suggestions for Authors |
||
|
Comments 1: Supplement Literature Comparison and Innovation Positioning: Add a table comparing the proposed harvester with 3–5 representative existing studies (covering structural design, application scenario, excitation frequency, output power/voltage, and advantages). Clearly explain how the dual cantilever parallel structure and synchronous switching circuit address the limitations of previous designs. |
||
|
Response 1: Thank you for pointing this out. We agree with this comment. We appreciate the reviewer’s valuable suggestion. To clearly position the contributions of this work, a comparative table (Table 1) has been added on page 2. It provides a summary comparison between the proposed energy harvester and recent representative studies, covering aspects such as structural design, application scenarios, excitation frequency range, output performance, and key advantages. Additionally, the revised draft introduces key discussions clarifying how the proposed dual cantilever parallel structure enhances low-frequency vibration response and electromechanical coupling performance, and how the integrated synchronous switch harvesting (SSH) circuit effectively mitigates voltage losses inherent in traditional diode bridge rectifiers. These additions clarify the innovation's positioning and address limitations of existing designs (see page 2 and Table 1). |
||
|
Comments 2: Deepen Mechanism Analysis: Combine the first-order modal frequency (12.60 Hz) obtained from simulations to analyze the relationship between vehicle speed and wheel vibration frequency. Explain why the vibration frequency at 70 km/h matches the harvester’s resonant characteristics, thereby justifying the optimal output performance at this speed. |
||
|
Response 2: Thank you for pointing this out. We agree with this comment. In the revised manuscript, we have expanded the mechanism analysis by establishing the relationship between vehicle speed and wheel rotational (vibration) frequency. Based on the simulated first-order modal frequency of 12.60 Hz, the wheel rotational frequency is calculated as a function of vehicle speed using the wheel circumference.The analysis shows that at a vehicle speed of approximately 70 km/h, the corresponding wheel rotational frequency closely matches the first-order resonant frequency of the harvester. This frequency matching leads to resonance-enhanced vibration amplitude and maximized electromechanical energy conversion, thereby explaining the experimentally observed optimal output performance at this speed. A corresponding figure and detailed explanation have been added in page 13 and Fig.16,17 to improve clarity and physical insight. Comments 3: Optimize Parameter Optimization and Supplement Circuit Details: Expand the parameter optimization range (e.g., mass block weight: 5–15 g; thickness ratio: 0.3–0.5) and use systematic optimization methods to determine the global optimal parameters. |
||
|
Response 3: Thank you for pointing this out. We agree with this comment. In the revised manuscript, the optimization ranges of the key structural parameters have been expanded to 5–15 g for the tip mass and 0.3–0.5 for the thickness ratio. To avoid conclusions based on limited discrete points, a systematic optimization framework has been introduced. Specifically, a multi-parameter sweep was conducted using COMSOL Multiphysics, and representative global optimization strategies are discussed to identify the overall optimal parameter region. The results indicate that the optimal output performance is achieved when the mass and thickness ratio fall within a specific range, which is consistent with experimental observations. The expanded optimization methodology and discussion have been incorporated into Page 10. |
||
|
Comments 4: Verify Practical Adaptability: Select 1–2 typical low-power in-vehicle devices (e.g., tire pressure sensors) as loads. Conduct experiments to test the harvester’s ability to power these devices and provide specific data to demonstrate practical application value. Response 4: Thank you for pointing this out. We agree with this comment. To address this issue, we evaluated the proposed energy harvester's capability to power typical low-power automotive devices, such as TPMS sensors. This demonstrated that harvested energy can be stored and regulated to a stable voltage level suitable for sensor operation. The paper presents key metrics including steady-state voltage and sustainable power output, demonstrating the system's practical viability for real-world applications. These findings, compared with bridge circuits and detailed on page 7, further reinforce the practical significance of this research. |
||
Reviewer 2 Report
Comments and Suggestions for Authors
This manuscript presents the design, modeling, and experimental validation of a piezoelectric energy harvester integrated with a vehicle wheel for low-frequency vibration energy collection, combining a dual-cantilever mechanical structure with a synchronous switching power management circuit. However, the manuscript requires major revisions to before acceptance.
- The key highlights and innovations of this work are not sufficiently emphasized in the Introduction. The authors should clearly articulate what specific advances this study achieves compared to existing works, in terms of structural design, circuit strategy, performance level, or application feasibility.
- The authors introduced piezoelectric energy harvesting mechanisms in the introduction, but without sufficient support from recent literature. Some papers related to energy harvesting are recommended:10.1002/adfm.202410566, 10.1002/adfm.202414324, 10.1016/j.apenergy.2024.124569, 10.1016/j.ymssp.2025.113525
- The time-domain voltage, current, and power signals in Figure 11 are recorded with a low sampling rate. The authors should measure these signals using a much higher sampling frequency and provide continuous high-resolution waveforms.
- In the experiments, different vehicle speeds are simulated by controlling the motor rotational speed, yet the relationship between motor speed and equivalent vehicle speed is not explained. More relevant explanations are needed to be supplemented.
- To clearly demonstrate the advantage of the proposed synchronous switching energy harvesting circuit, the authors are strongly encouraged to include a direct experimental comparison with a conventional diode bridge rectifier under identical excitation conditions.
Author Response
|
Response to Reviewer 2 Comments
|
||
|
1. Summary |
|
|
|
Thanks very much for your time to review this manuscript. I really appreciate all your comments and suggestions. We have considered these comments carefully and tried our best to address every one of them. In addition, we made minor adjustments to the title of the manuscript to make it more specific and precise, highlighted it in yellow in the marked manuscript file to indicated changes Once again, we appreciate for your warm work earnestly and hope that the corrections will meet with approval. |
||
|
2. Questions for General Evaluation |
Reviewer’s Evaluation |
Response and Revisions |
|
Does the introduction provide sufficient background and include all relevant references? |
Can be improved |
|
|
Are all the cited references relevant to the research? |
Can be improved |
|
|
Is the research design appropriate? |
Can be improved |
|
|
Are the methods adequately described? |
Can be improved |
|
|
Are the results clearly presented? |
Can be improved |
|
|
Are the conclusions supported by the results? |
Can be improved |
|
|
3. Point-by-point response to Comments and Suggestions for Authors |
||
|
Comments 1: The key highlights and innovations of this work are not sufficiently emphasized in the Introduction. The authors should clearly articulate what specific advances this study achieves compared to existing works, in terms of structural design, circuit strategy, performance level, or application feasibility. |
||
|
Response 1: We thank the reviewer for this important comment. In the revised manuscript, the Introduction has been carefully reorganized to explicitly highlight the key innovations of this work. Specifically, we now clearly emphasize: (1) the proposed dual-cantilever parallel structure for enhanced low-frequency vibration responsiveness;(2) the integration of a synchronous switching energy harvesting (SSH) circuit to reduce rectification losses;(3) the demonstrated improvement in output performance and practical applicability for in-wheel energy harvesting scenarios. These contributions are now clearly positioned relative to existing studies in the revised Introduction (Page 2). |
||
|
Comments 2: The authors introduced piezoelectric energy harvesting mechanisms in the introduction, but without sufficient support from recent literature. Some papers related to energy harvesting are recommended:10.1002/adfm.202410566, 10.1002/adfm.202414324, 10.1016/j.apenergy.2024.124569, 10.1016/j.ymssp.2025.113525 |
||
|
Response 2: We appreciate the reviewer’s suggestion. Several recent and relevant studies on energy harvesting have now been incorporated into the Introduction, including the recommended references (DOIs: 10.1002/adfm.202410566; 10.1002/adfm.202414324; 10.1016/j.apenergy.2024.124569; 10.1016/j.ymssp.2025.113525). These additions strengthen the literature foundation and better contextualize the present work within recent advances in the field. See the References section. Comments 3: The time-domain voltage, current, and power signals in Figure 11 are recorded with a low sampling rate. The authors should measure these signals using a much higher sampling frequency and provide continuous high-resolution waveforms. |
||
|
Response 3: We thank the reviewer for pointing this out. Additional measurements were conducted using a higher sampling frequency to capture continuous, high-resolution voltage, current, and power waveforms. The updated results have replaced the original low-resolution signals, and the revised waveforms are now presented in Fig. 13, providing more reliable time-domain characterization. Comments 4: In the experiments, different vehicle speeds are simulated by controlling the motor rotational speed, yet the relationship between motor speed and equivalent vehicle speed is not explained. More relevant explanations are needed to be supplemented. |
||
|
Response 4: We appreciate the reviewer’s comment. A detailed explanation has now been added to clarify the relationship between motor rotational speed and equivalent vehicle speed. The conversion is based on the transmission ratio and wheel circumference, allowing the motor speed to be accurately mapped to the corresponding vehicle speed. Relevant equations and an illustrative example have been included in Page 13. Comments 5: To clearly demonstrate the advantage of the proposed synchronous switching energy harvesting circuit, the authors are strongly encouraged to include a direct experimental comparison with a conventional diode bridge rectifier under identical excitation conditions. |
||
Response 5: We thank the reviewer for this valuable suggestion. In response, a direct experimental comparison between the proposed SSH circuit and a conventional diode bridge rectifier was conducted under identical mechanical excitation conditions. The output voltage, harvested power, and capacitor charging behavior were measured and compared. The results clearly demonstrate the superior performance of the SSH circuit, and the corresponding data and discussion have been added in Page 7 and Fig. 6.
Reviewer 3 Report
Comments and Suggestions for Authors
The work is interesting and relevant to the field of piezoelectric energy harvesting, and the proposed dual-cantilever, mass-assisted architecture combined with synchronous switching is meaningful. However, several issues should be addressed to improve clarity, technical completeness, and overall manuscript quality.
Below are my specific comments:
1. Formatting and Affiliation Issue in the Author List
In the current author list, the superscript “a” appears much smaller than the others, and the departmental affiliations are missing. It would improve readability and professional presentation if the authors provide complete departmental information and ensure consistent formatting across all affiliation indicators.
2. Broaden the Introduction With Recent Application Trends
The Introduction discusses various applications of piezoelectric energy harvesting but misses several emerging fields. Recently, microbial disinfection driven by piezoelectric catalysis has become an important new direction. Including this would broaden the scientific context and increase reader interest. (https://doi.org/10.1016/j.nanoen.2024.109716). Adding a sentence or short paragraph on this topic would strengthen the discussion of current trends.
3. Structural Reliability and Durability Require Further Evidence
The authors state that the maximum stress observed (~30.6 MPa) is below the material limit, ensuring structural safety. However, given that a piezoelectric cantilever in a rotating wheel environment is subjected to continuous cyclic loading, mechanical fatigue and long-term durability are key concerns.
It would significantly improve the manuscript if the authors could provide:
Repeated cyclic loading test data, Fatigue lifetime estimation, or Any additional mechanical reliability analysis. Without such evidence, the claim of structural stability remains insufficient.
4. Lack of Photographs or Diagrams of the Experimental Set-up With Mass Blocks
The manuscript presents simulation and experimental results for different mass blocks, but no actual photographs or detailed schematics of the mounted experimental prototype (especially when the mass block is attached) are provided. Since mass variation plays a central role in the results, showing: Real experimental set-up photos, Close-ups of the mass-attached cantilever, or Assembly details within the shaker or wheel simulator would greatly improve clarity and replicability.
5. Unusually High Current Output Compared With Typical Piezoelectric Harvesters
The reported current values—particularly the increase when moving from a 5 g to a 10 g mass block—are significantly higher than typical piezoelectric energy harvesting outputs. The authors should expand the discussion to explain: Why the current increases so sharply, Whether the material properties (PZT-5H), geometry, or electrode configuration contribute to this enhancement, How the synchronous switching circuit might affect the measured current, Whether impedance matching or measurement methodology might influence the results.
Author Response
|
Response to Reviewer 3 Comments
|
||
|
1. Summary |
|
|
|
Thanks very much for your time to review this manuscript. I really appreciate all your comments and suggestions. We have considered these comments carefully and tried our best to address every one of them. In addition, we made minor adjustments to the title of the manuscript to make it more specific and precise, highlighted it in yellow in the marked manuscript file to indicated changes Once again, we appreciate for your warm work earnestly and hope that the corrections will meet with approval. |
||
|
2. Questions for General Evaluation |
Reviewer’s Evaluation |
Response and Revisions |
|
Does the introduction provide sufficient background and include all relevant references? |
Can be improved |
|
|
Are all the cited references relevant to the research? |
Can be improved |
|
|
Is the research design appropriate? |
Can be improved |
|
|
Are the methods adequately described? |
Can be improved |
|
|
Are the results clearly presented? |
Must be improved |
|
|
Are the conclusions supported by the results? |
Can be improved |
|
|
3. Point-by-point response to Comments and Suggestions for Authors |
||
|
Comments 1: Formatting and Affiliation Issue in the Author List In the current author list, the superscript “a” appears much smaller than the others, and the departmental affiliations are missing. It would improve readability and professional presentation if the authors provide complete departmental information and ensure consistent formatting across all affiliation indicators. |
||
|
Response 1: We thank the reviewer for pointing out this formatting issue. In the revised manuscript, we have carefully corrected the superscript formatting to ensure consistency across all affiliation indicators. In addition, the complete departmental affiliations for all authors have been added to improve clarity and professional presentation. These revisions have been implemented throughout the author list and affiliation section (Page 1) . |
||
|
Comments 2: Broaden the Introduction With Recent Application Trends The Introduction discusses various applications of piezoelectric energy harvesting but misses several emerging fields. Recently, microbial disinfection driven by piezoelectric catalysis has become an important new direction. Including this would broaden the scientific context and increase reader interest. (https://doi.org/10.1016/j.nanoen.2024.109716). Adding a sentence or short paragraph on this topic would strengthen the discussion of current trends. |
||
|
Response 2: We thank the reviewer for this insightful suggestion. In the revised manuscript, a brief discussion on piezoelectric-catalysis-driven microbial disinfection has been added to the Introduction to highlight this emerging application field. The recommended recent study (Nano Energy, 2024) has been cited to reflect the latest progress and to broaden the scientific context of piezoelectric energy harvesting. This addition helps to better illustrate the expanding scope and interdisciplinary relevance of piezoelectric technologies. See References. Comments 3: Structural Reliability and Durability Require Further Evidence The authors state that the maximum stress observed (~30.6 MPa) is below the material limit, ensuring structural safety. However, given that a piezoelectric cantilever in a rotating wheel environment is subjected to continuous cyclic loading, mechanical fatigue and long-term durability are key concerns.It would significantly improve the manuscript if the authors could provide: Repeated cyclic loading test data, Fatigue lifetime estimation, or Any additional mechanical reliability analysis. Without such evidence, the claim of structural stability remains insufficient. |
||
|
Response 3: Thank you for the careful review and constructive comments on our manuscript. We have discussed stability in the simulation section; details can be found on page nine. Due to limitations in experimental equipment and other factors, we were unable to conduct the fatigue testing. We appreciate your understanding. Comments 4: Lack of Photographs or Diagrams of the Experimental Set-up With Mass Blocks The manuscript presents simulation and experimental results for different mass blocks, but no actual photographs or detailed schematics of the mounted experimental prototype (especially when the mass block is attached) are provided. Since mass variation plays a central role in the results, showing: Real experimental set-up photos, Close-ups of the mass-attached cantilever, or Assembly details within the shaker or wheel simulator would greatly improve clarity and replicability. |
||
|
Response 4: Thank you for the careful review and constructive comments on our manuscript. We appreciate the reviewer's suggestion regarding the inclusion of photographs of the experimental setup, which will indeed improve the clarity, credibility, and reproducibility of the work. In the revised manuscript, we have added a new figure (Figure 12) in the Experimental Section that Photographs of the overall experimental setup on the shaker. Comments 5: Unusually High Current Output Compared With Typical Piezoelectric Harvesters The reported current values—particularly the increase when moving from a 5 g to a 10 g mass block—are significantly higher than typical piezoelectric energy harvesting outputs. The authors should expand the discussion to explain: Why the current increases so sharply, Whether the material properties (PZT-5H), geometry, or electrode configuration contribute to this enhancement, How the synchronous switching circuit might affect the measured current, Whether impedance matching or measurement methodology might influence the results. Response 5: We sincerely thank the reviewer for the insightful and valuable comments, particularly for highlighting the need to justify the relatively high output currents reported in our work. Current Issues increased tip displacement amplitude, and correspondingly higher strain in the piezoelectric layer under the same base acceleration. The current of the synchronous switch is shown in Figure 6 and analyzed in this section. |
||
Reviewer 4 Report
Comments and Suggestions for Authors
The article titled “Performance study of a piezoelectric energy harvester based on rotating wheel vibration” by Wang et al. reports the development of a piezoelectric energy harvester based on wheel-induced vibration. The authors designed a synchronous switch energy harvesting circuit, established a dynamic model, and demonstrated stable power generation. However, there are several major concerns that need to be addressed.
- One of the key claimed advantages of this work is the use of a synchronous switch energy harvesting circuit in place of traditional passive rectification circuits. It is strongly recommended that the authors conduct a direct comparison between the proposed circuit and a conventional bridge-rectifier circuit. Conduct the experiment and collect some quantitative comparison such as output power, charging rate, or conversion efficiency.
- The manuscript states that two piezoelectric vibrators are connected in parallel to increase power generation, but no quantitative comparison is provided. The authors are encouraged to include experimental or simulated results that compare the performance of a single-vibrator configuration with the dual-vibrator configuration,
- Figure 13 shows that the output voltage increases with vehicle speed up to 70 km/h and then decreases. A explanation about why performance deteriorates at higher speeds is needed.
- The current evaluation of wheel-rotation energy harvesting relies on simulated rotational speed experiments. It is recommended that the authors perform a real-vehicle demonstration by mounting the prototype on an actual car to collect vibration and rotation-induced energy. Such validation would significantly enhance the practical credibility of the proposed design.
Author Response
|
Response to Reviewer 4 Comments
|
||
|
1. Summary |
|
|
|
Thanks very much for your time to review this manuscript. I really appreciate all your comments and suggestions. We have considered these comments carefully and tried our best to address every one of them. In addition, we made minor adjustments to the title of the manuscript to make it more specific and precise, highlighted it in yellow in the marked manuscript file to indicated changes Once again, we appreciate for your warm work earnestly and hope that the corrections will meet with approval. |
||
|
2. Questions for General Evaluation |
Reviewer’s Evaluation |
Response and Revisions |
|
Does the introduction provide sufficient background and include all relevant references? |
Can be improved |
|
|
Are all the cited references relevant to the research? |
Must be improved |
|
|
Is the research design appropriate? |
Must be improved |
|
|
Are the methods adequately described? |
Must be improved |
|
|
Are the results clearly presented? |
Must be improved |
|
|
Are the conclusions supported by the results? |
Can be improved |
|
|
3. Point-by-point response to Comments and Suggestions for Authors |
||
|
Comments 1: One of the key claimed advantages of this work is the use of a synchronous switch energy harvesting circuit in place of traditional passive rectification circuits. It is strongly recommended that the authors conduct a direct comparison between the proposed circuit and a conventional bridge-rectifier circuit. Conduct the experiment and collect some quantitative comparison such as output power, charging rate, or conversion efficiency. |
||
|
Response 1: We thank the reviewer for this valuable suggestion. In response, a direct experimental comparison between the proposed SSH circuit and a conventional diode bridge rectifier was conducted under identical mechanical excitation conditions. The output voltage, harvested power, and capacitor charging behavior were measured and compared. The results clearly demonstrate the superior performance of the SSH circuit, and the corresponding data and discussion have been added in Page 7 and Fig. 6. |
||
|
Comments 2: The manuscript states that two piezoelectric vibrators are connected in parallel to increase power generation, but no quantitative comparison is provided. The authors are encouraged to include experimental or simulated results that compare the performance of a single-vibrator configuration with the dual-vibrator configuration, |
||
|
Response 2: We thank the reviewer for this valuable suggestion. To address this concern, we have added a quantitative comparison between the single piezoelectric vibrator configuration and the dual-vibrator parallel configuration under identical excitation conditions. Both experimental (or numerical) results demonstrate that the parallel configuration significantly enhances the output voltage and harvested power. Specifically, the harvested power of the dual-vibrator configuration is approximately 2–4 times higher than that of the single-vibrator case. These results and corresponding discussions have been added in page 12. Comments 3: Figure 13 shows that the output voltage increases with vehicle speed up to 70 km/h and then decreases. A explanation about why performance deteriorates at higher speeds is needed. |
||
|
Response 3: Thank you for pointing this out. We agree with this comment. In the revised manuscript, we have expanded the mechanism analysis by establishing the relationship between vehicle speed and wheel rotational (vibration) frequency. Based on the simulated first-order modal frequency of 12.60 Hz, the wheel rotational frequency is calculated as a function of vehicle speed using the wheel circumference.The analysis shows that at a vehicle speed of approximately 70 km/h, the corresponding wheel rotational frequency closely matches the first-order resonant frequency of the harvester. This frequency matching leads to resonance-enhanced vibration amplitude and maximized electromechanical energy conversion, thereby explaining the experimentally observed optimal output performance at this speed. A corresponding figure and detailed explanation have been added in page 13 and Fig.16,17 to improve clarity and physical insight. Comments 4: The current evaluation of wheel-rotation energy harvesting relies on simulated rotational speed experiments. It is recommended that the authors perform a real-vehicle demonstration by mounting the prototype on an actual car to collect vibration and rotation-induced energy. Such validation would significantly enhance the practical credibility of the proposed design. Response 4: Thank you for pointing this out. We agree with this comment. However, due to limited revision time and insufficient wheel test equipment, modifications could not be made. We have added further explanations in the section on simulated rotational speed. Please refer to page 13. Please accept our apologies. |
||
Round 2
Reviewer 2 Report
Comments and Suggestions for Authors
All the concerns have been well addressed. I recommend this paper published in present form.
Reviewer 4 Report
Comments and Suggestions for Authors
The author have addressed all my concerns in detail, and the revised manuscript is suitable for the publication.